# Copper Surface Treatment Method with Antibacterial Performance Using “Super-Spread Wetting” Properties

**DOI:** 10.3390/ma15010392

**Published:** 2022-01-05

**Authors:** Beomdeok Seo, Hideyuki Kanematsu, Masashi Nakamoto, Yoshitsugu Miyabayashi, Toshihiro Tanaka

**Affiliations:** 1Division of Materials and Manufacturing Science, Graduate School of Engineering, Osaka University, 2-1 Yamadaoka, Suita 565-0871, Osaka, Japan; nakamoto@mat.eng.osaka-u.ac.jp (M.N.); tanaka@mat.eng.osaka-u.ac.jp (T.T.); 2Department of Materials Science and Engineering, National Institute of Technology (KOSEN), Suzuka College, Suzuka 510-0294, Mie, Japan; kanemats@mse.suzuka-ct.ac.jp; 3Graduate School of Engineering, Osaka University, 2-8 Yamadaoka, Suita 565-0871, Osaka, Japan; miyabayashi@jrl.eng.osaka-u.ac.jp

**Keywords:** antibacterial properties, coating, copper, fine crevice structure, super-spread wetting properties

## Abstract

In this work, a copper coating is developed on a carbon steel substrate by exploiting the superwetting properties of liquid copper. We characterize the surface morphology, chemical composition, roughness, wettability, ability to release a copper ion from surfaces, and antibacterial efficacy (against *Escherichia coli* and *Staphylococcus aureus*). The coating shows a dense microstructure and good adhesion, with thicknesses of approximately 20–40 µm. X-ray diffraction (XRD) analysis reveals that the coated surface structure is composed of Cu, Cu_2_O, and CuO. The surface roughness and contact angle measurements suggest that the copper coating is rougher and more hydrophobic than the substrate. Inductively coupled plasma atomic emission spectroscopy (ICP-AES) measurements reveal a dissolution of copper ions in chloride-containing environments. The antibacterial test shows that the copper coating achieves a 99.99% reduction of *E. coli* and *S. aureus*. This study suggests that the characteristics of the copper-coated surface, including the chemical composition, high surface roughness, good wettability, and ability for copper ion release, may result in surfaces with antibacterial properties.

## 1. Introduction

The Centers for Disease Control and Prevention reported that healthcare-associated infections (HAIs) cause or contribute to 99,000 deaths and add approximately $40 billion to healthcare coasts each year [1]. As a possible cause of infection, bacterial contamination on the surfaces of materials, especially in hospitals and public places, is proposed as a serious threat [2]. On surfaces, many types of bacteria can survive for long periods, with some even able to survive for more than a month [2]. Various efforts, such as hand washing, disinfection, and antibacterial surfaces, have been developed to control infection, but the problem has not been resolved [2,3,4]. A recent trend in risk management of the transfer of bacteria from surface to surface is the use of copper in the manufacture of public and hospital materials [5,6]. Some of these studies reported that the use of copper alloys in intensive care unit rooms can significantly reduce HAIs compared with a standard room [5,7].

Although the antibacterial mechanism of the solid copper surface has yet to be clearly understood, several studies have investigated the result of the so-called contact killing [8,9]. When bacteria are directly in contact with metallic copper, copper ions accumulate inside the cell because the bacteria recognize the copper ions as essential nutrients [10,11]. The cell and DNA are then damaged and destroyed by the depolarization effect and reactive oxygen species (ROS) [12]. Surface properties, including roughness, wetting behavior, and contact angle, significantly influence contact killing [12,13,14,15,16,17,18,19,20]. Together with the contact-killing effect, the direct release of copper ions from metallic copper plays a decisive role in the bacterial killing process [21,22,23]. Copper ions prompt the generation of ROS and cause bacterial cell damage or death. It is also reported that there are differences depending on the type of copper oxide. The antibacterial performance can be further improved by applying a copper oxide surface because of the extensive release of copper ions from the copper oxide surface [23,24,25]. Based on these observations, the use of copper is a promising strategy to prevent HAIs.

When applying copper to materials in public and hospital settings, it is the preferred method to coat copper on an inexpensive metal such as carbon steel in consideration of the economic aspects [26]. However, there are problems associated with the copper coating process. The widely used copper-plating process uses cyanide ions, which can cause serious environmental pollution problems [27,28]. Other methods, such as plasma treatment with oxygen, chemical vapor deposition, and ammonia plasma, require complex equipment and procedures [29,30,31,32]. Therefore, to widely apply the antibacterial properties of copper, a new method to solve these problems is important.

In this study, for copper surface coating, the super-spread wettability properties of liquid copper are exploited. The literature reports that liquid copper is not able to wet a solid oxide [33,34]. However, our previous works have shown that liquid copper unusually penetrates and spreads on a surface with fine crevice structures formed by capillary action [35]. The resulting phenomenon, named “super-spread wetting”, caused by the capillary characteristics of the liquid metal and metal surface with a fine crevice structure, differs from an ordinary occurrence [36,37,38]. In addition, using the super-spread wetting property, the liquid copper is able to flow to the desired target point [39]. Although some research has been conducted to understand this unusual phenomenon, studies on the coating technology have not been conducted [40,41].

We provide a new method of copper coating with antibacterial properties on carbon steel using the super-spread wetting properties of liquid copper. In addition, our research suggests a method for coating copper that does not use complex manufacturing equipment and processes and does not emit pollutants. The surface characterizations, including morphology, chemical composition, phase, roughness, wettability, and the ability for a copper ion to be released from the surfaces, have been systematically investigated. Furthermore, the antibacterial properties are determined by ISO 22196:2011 method against *Escherichia coli* and *Staphylococcus aureus*. In this work, we will discuss the antibacterial mechanism for copper-coated surfaces.

## 2. Materials and Methods

### 2.1. Materials and Fabrication of the Test Samples

The substrate was cut from JIS-SS400 carbon steel plate with the chemical composition: C 0.148; Si 0.213; Mn 0.458; S 0.018; P 0.012 (wt.%), Fe balance. The specimens were machined into rectangular shapes, with dimensions of 10 mm × 10 mm × 2 mm, then sequentially ground by emery papers up to 1200 grit and degreased in acetone using an ultrasonic bath. Copper powder (99 purity, Sigma Aldrich, St Louis, MO, USA) was applied for the coating process.

Figure 1 shows the different steps involved with fabricating the test samples. First, a fine structure was formed on the surface to allow liquid copper to spread on the surface. Our previous research confirmed that surfaces with a fine crevice structure can be created by laser irradiation [38]. As shown in Figure 1a, a continuous Nd: YAG laser (ML-7062A, Miyachi Corporation, Tokyo, Japan) was used to fabricate the fine crevice structure with two types of patterns: covering all (10 × 10 mm square) and 48% (0.8 mm × 9.9 mm × 6 pcs rectangle arranged at intervals of 1.0 mm) of the substrate, respectively. Laser irradiation was performed on the substrate positioned 110 mm under the scanning lens with an average power of 30 W at a frequency of 6.0 kHz, a spot diameter of 0.1 mm, a pitch of 0.01 mm and a scan speed of 9.0 mm/s under air atmosphere. Then, the prepared substrate with a crevice structure was fed with 6 mg/cm^2^ copper and heated to 1100 °C, which is slightly above the melting point of copper, to coat it with liquid copper in an electric furnace (see Figure 1b) [36]. The temperature profile for the coating process is shown in Figure 1c. To prevent oxidation during the heating process, Ar gas (60 mL/min) and H_2_ gas (15 mL/min) were supplied, and the oxygen partial pressure was maintained at approximately 10^−3^ atm in the furnace. After a prescribed heating time, the specimens were cooled in the furnace at a rate of 400 °C per hour to 600 °C under an Ar gas environment and then air-cooled. The cooling condition was determined by focusing on the formation of copper and copper compounds with antibacterial effects by using a thermodynamic calculation with the FactSage software (version 7.1).

### 2.2. Strains and Culture Conditions

We used two different bacterial strains to determine the antibacterial effect, *Escherichia coli* (*E. coli*, ATCC 25922) and *Staphylococcus aureus* (*S. aureus*, ATCC12228), the most frequently used gram-negative and gram-positive organisms, respectively. All the investigations were conducted in Luria Broth (LB, Nacalai tesque, Kyoto, Japan) consisting of 10 g of bactotrypton, 5 g of yeast extract and 10 g of NaCl per liter. All solutions were sterilized by autoclaving at 120 °C for 15 min before use. Both bacteria were cultured in 10 mL of LB broth on a swing bed at 35 °C overnight. This solution was then diluted in sterile LB broth to 10^5^ CFU/mL.

### 2.3. Surface Characterization

The surface morphology and composition were investigated using scanning electron microscopy (SEM, Miniscope TM-1000, Hitachi, Japan) and energy-dispersive spectroscopy (EDS, JSM-6500F, JEOL, Tokyo, Japan). The surface phase was analyzed with X-ray diffraction analysis (XRD, SmartLab, Rigaku, Tokyo, Japan) using CuKα radiation in the range of 2θ from 20°–80°. The surface roughness and profile were measured by three-dimensional (3D) laser scanning microscopy (VK-9700, Keyence, Osaka, Japan) and analyzed with the VK-H1XP software. Results for five random areas were presented as the average roughness (R_a_), peak-to-valley roughness (R_z_) and root-mean-square roughness (R_q_). The contact angle measurement was used to characterize the wettability by LB broth. The contact angle was measured at room temperature using the sessile drop method from a contact angle meter (CA, DMo-501, Kyowa Interface Science, Saitama, Japan). The measurements were repeated three times for each specimen. The “Standard test methods for measuring adhesion by tape test (ASTM D3359-02)” were performed to investigate the adhesion of the copper coating on the specimens [42].

### 2.4. Anti-Bacterial Activity Test

We used a modified ISO 22196:2011 (Measurement of Antibacterial Activity on Plastics and Other Non-Porous Surfaces) test to characterize the antibacterial properties of the specimens [43,44]. Prior to the experiments, the specimens were sterilized with 75 vol.% ethyl alcohol and UV-light for 30 min. The prepared bacterial suspension (16 µL) was applied to the specimen surfaces. A piece of polymer film, 10 × 10 mm, sterilized with 70% ethanol and dried, was placed on the surface to spread the suspension and to reduce evaporation. The samples were then incubated for 24 h at 35 °C. After the incubation, the specimen and polymer film were vortexed for 1 min with 1 mL of sterilized water containing Tween 80 (20 µL) to remove attached bacteria from the surfaces. The bacterial solution was diluted with fresh LB broth by a factor of 10–10^4^. Subsequently, 100 µL of diluted solution was evenly distributed over the surface of the LB agar in petri dishes, followed by incubation at 35 °C for 24 h. Afterwards, the numbers of bacterial colonies were counted to determine the bacterial cell concentration. Each sample type was tested in triplicate. Log reduction was determined by the following equation (Equation (1)) [45,46]:(1)$$\mathrm{Log}\;\mathrm{reduction}=\log_{10}\frac{A}{B}$$
where *A* and *B* refer to the number of bacterial colonies on the control sample and test sample, respectively, after a designated contact time.

### 2.5. Measurement of Copper Ion Release

The copper ion release from the coating was measured using inductively coupled plasma-atomic emission spectrometry (ICP–AES, Optima 8300, Norwalk, Connecticut, USA). The specimens were immersed in 100 mL of sterilized LB broth, and then, the solution was extracted from samples after 1, 4, 8, and 24 h to analyze for copper release. The copper ion concentration was quantified with a standard ICP ionic solution with different concentrations (from 0.02 to 0.5 ppm) that was used to plot the calibration curve. The element copper was analyzed using an emission wavelength of 325 nm.

## 3. Results

### 3.1. Characteristics of Copper Coating by Super-Wetting Properties

#### 3.1.1. Surface Morphology and Cross-Sectional Analysis

Figure 2 indicates the scanning electron microscopy (SEM) images before and after coating. The substrate had a surface with a smooth and even structure (Figure 2a). Our previous work confirmed that laser irradiation melts the metal and causes swelling and spattering, and as a result, the liquid metal accumulates and forms a fine crevice structure (Figure 2b) [38]. Before the coating process, the fine crevice structure formed on all and 48% of the substrate by laser irradiation (Figure 2c,d). After the coating process by the super-spread wetting properties, the copper was coated on the surface (Figure 2e,f). It clearly indicates that the coating film was formed only on the fine crevice structure fabricated by laser irradiation (Figure 2f). This means that we are able to coat copper on the required area through the process of modifying the surface.

Figure 3 shows the cross-sectional images and the energy-dispersive X-ray spectroscopy (EDS) results for copper coating by the superspreading properties. The cross-sectional images reveal that the coating, with approximately 20–40 µm thickness and homogeneous microstructure, was composed of a substrate outer layer. Furthermore, the copper-coating surface had the highest adhesion strength grade of 5B (no detachment of the squares of the lattice), according to ASTM D3359-02, as shown in Table 1 [42]. Furthermore, it clearly shows that copper was coated along a complex surface structure, as shown in Figure 3e,f. This phenomenon results from liquid copper penetrating and spreading through the complex fine crevice structure by capillary action [35,36]. These results provide evidence that a coating can be controlled by the fine crevice surface structure and super-spread wetting properties. Furthermore, additional research is needed for the possibility of controlling the coating thickness according to the surface structure and the copper supplied.

#### 3.1.2. Phase Analysis

Figure 4 shows the XRD patterns of the substrate and the copper-coated specimens by the super-spread wetting properties in the range of 20°–80°. According to the XRD analysis results of the all- and 48%-copper-coated surfaces, diffraction peaks were observed at 2θ values of 43.39°, 50.49°, and 74.18°, which corresponded to the (111), (200) and (200) planes of the metallic Cu (JCPDS No. 04-0836). In addition to the peaks of metallic Cu, the diffraction peaks at 36.4° and 42.3° corresponded to the (111) and (200) planes of the Cu_2_O (JCPDS No. 05-0667), and those at 33.17°, 35.4°, and 38.7° corresponded to (100), (002), and (111) planes of CuO (JCPDS No. 48-1548). Of course, peaks from the crystal phases of α-Fe and Fe_3_O_4_ related to the substrate also appeared. These results indicate that the substrate with a surface crevice structure was coated with Cu by the super-spread wetting properties. In addition, as predicted by the thermodynamics calculation, the surface structure formed from Cu to Cu, Cu_2_O and CuO by oxidation during the fabrication and cooling processes.

#### 3.1.3. Measurement of Surface Roughness and Wettability

To further identify the surface topology after the coating process by the super-spread wetting properties, the surface roughness was assessed by a 3D laser scanning microscope, as shown in Figure 5. The substrate showed a homogeneous and regular topology, with low peaks (green) and low valleys (yellow) (Figure 5a,c). In contrast, the mountains (red) and valleys (blue) identified on the coated surface revealed a heterogeneous and irregular topology (Figure 5b,d). Table 2 offers, in addition, some objective parameters to determine the surface characteristics of the samples, i.e., average roughness (R_a_), peak-to-valley roughness (R_z_) and root-mean-square roughness (R_q_). Based on statistical analysis, the difference before and after the coating process was obvious. For example, R_a_ was 0.24 and 6.35 µm before and after coating, respectively. The R_z_ and R_q_ parameters, reflecting the local height variations in a surface area, were 9.55 and 0.33 µm before and 80.57 and 7.89 µm after coating, respectively. This result is caused by the super-spread wetting properties of liquid copper through capillary action into the surface with crevice structure fabricated by a laser.

Contact angle measurements characterized the degree of wettability for the specimens in the LB broth, and representative images and average values are shown in Figure 6. The substrate before the coating process had a contact angle of 61.3° (Figure 6a). After the coating process, the specimen became more wettable, which was indicated by a reduction of the contact angle to 56.5° (Figure 6b). Therefore, the coating process by the super- spread wetting properties led to changes in the wettability (Figure 6c).

### 3.2. Anti-Bacterial Nature of the Cu Coating by Super-Spread Wetting Properties

The antibacterial nature of the specimens coated by the super-spread wetting properties was evaluated by ISO 22196:2011 against *E. coli* and *S. aureus*. Figure 7a shows representative images of bacteria colonies after incubation for *E. coli* and *S. aureus* of the solution collected after contact with the specimen for 24 h. Large amounts of bacterial colonies appeared on the substrate (control) for both types of bacteria. In contrast, in the dish cases, for the all copper-coated samples that were fabricated with the super-spread properties, there were no colonies for both types of bacteria and a similar aspect as the copper plate was even shown. As shown in Figure 7b, the number of bacteria inoculated on the copper-coated surface was reduced to less than 10 CFU/mL after a 24-h incubation period. Therefore, the log reduction value for all copper-coated samples against *E. coli* and *S. aureus* was 4.08 and 4.08, respectively, by Equation (1) (see Figure 7c). The copper-coated surface supported less than 0.01% of both types of bacterial grown on the copper coating by super-spread wetting properties. These results demonstrate that the copper-coated specimens by super-spread wettability have antibacterial properties to both gram-positive and gram-negative bacteria. In addition, on the 48%-copper-coated samples, only a few bacterial colonies were found. The number of bacteria inoculated on the 48%-copper-coated surface was reduced to (2.1 ± 0.1) × 10^3^ CFU/mL of *E. coli* and (2.6 ± 0.1) × 10^3^ CFU/mL of *S. aureus* after a 24-h incubation period (see Figure 7b). 

### 3.3. Copper-Ion Release

Figure 8 summarizes the release of copper ions from the all- and 48%-copper-coated samples versus exposure time in uninoculated LB broth. The copper ion concentration for the all-copper-coated sample increased with the extension of immersion duration time. However, copper emission from the 48%-copper-coated sample proceeded rapidly in the early immersion stage but was relatively slower over time. This was proposed to arise from the difference of the chemical state depending on the exposed area of the copper and the oxide type. After 24 h, the concentration of copper ions was 295 ppb and 100 ppb for the all- and 48%-copper-coated samples, respectively. These data clearly showed that copper ions were released from both specimens.

## 4. Discussion

Figure 9 presents why the copper coating with the super-spread wetting property is antibacterial, based on the experimental results so far. The coated specimen has an irregular surface structure because liquid copper is wetted to the crevice structure by capillary action using the superwetting property. Several studies found that these micro-sized rough surfaces enhance bacterial adhesion to the surface, described as an anchoring effect [20,47]. Bacteria would contact the rough surface easily composed of copper and copper oxides that have antibacterial performance formed through the coating process. Additionally, the coated surfaces are 56.5° through contact angle measurement, which means that the surface has a hydrophilic character with good wettability. It is known through many studies that when the surface has hydrophilic properties, bacteria can easily attach to the surface [15,16]. These wetting properties can further improve the antibacterial properties of copper compounds known as contact-killing. The influx of copper ions into the cytoplasm would be the key to antibacterial performance in contact-killing [48,49]. In addition, the antimicrobial efficacy of a copper coating is dependent on the number of copper ions released from the surface to the electrolyte. The LB broth used in this study contains chloride, and copper has the property of being dissolved in the form of complex ions in such an environment [50,51,52]. This phenomenon is related to the breaking of the equilibrium state of the copper surface into a polarized state by chloride. The elution phenomenon of copper from the coating surface can be explained as follows. When the coated surface is exposed to these environments, transitional products (CuCl_ads_) are formed according to the interaction between copper atoms on the coating surface and Cl. Because this product is not stable, it combines with more Cl^−^ ions and oxidizes into soluble oxidation products CuCl_2_^−^, as shown in Equations (2)–(4) [50,51,52].
(2)$$\mathrm{Cu}+\to {\;\mathrm{CuCl}}_{\mathrm{ads}}^{-}$$
(3)$${\mathrm{CuCl}}_{\mathrm{ads}}^{-}{+\mathrm{Cl}}^{-}\;\to {\;\mathrm{CuCl}}_{2}^{-}$$
(4)$${\mathrm{Cu}+2\mathrm{Cl}}^{-}\;\to {\;\mathrm{CuCl}}_{2}^{-}+2e^{-}$$

It has been reported that when the copper ion concentration in the immersion solution is higher than 0.036 mg/L, the antibacterial rate is more than 99% [21,22]. Our results also indicate that the copper ions concentration in the immersion solution is more than 295 and 110 ppb in the all- and 48%-copper-coated specimens, respectively. For that reason, it is considered that antibacterial properties are shown not only in the specimen coated with copper over the entire area, but also in the specimen in which the base material is partially exposed. The toxicity of copper ions is still unclear but is usually owing to their ability to catalyze Fenton chemistry according to Equation (5) [8,53]. Combined with Equation (6), these reactions can provide a reactive oxygen species that can destroy bacterial cells [53].
(5)$${\mathrm{Cu}}^{+}{+H}_{2}O_{2}\;\to {\;\mathrm{Cu}}^{2+}+{\mathrm{OH}}^{-}+{\mathrm{OH}}^{∙}$$
(6)$$H_{2}O_{2}+{\mathrm{OH}}^{∙}\;\to {\;H}_{2}O+O_{2}{}^{-}+H^{+}$$

## 5. Conclusions

Using the superwetting property of liquid copper, we have coated copper on carbon steel surfaces and also only on a desired area. In addition, we have discussed the properties of the coating surface and the correlation between antibacterial properties. The results demonstrated that a coating without visible cracks or voids between two metals can be manufactured using the superwetting property of liquid copper. In addition, we confirm that coating using the superwetting property has excellent antibacterial performance. Additionally, it is also interesting that it has excellent antibacterial performance even when coated over only 48% of the area. It is considered that the properties of the copper coating, which include the chemical composition of the surface, high surface roughness, good wettability, and ability for copper ion release, influence the antibacterial properties. Our research presents a new method for copper coating with antibacterial properties in a simple to produce and environmentally friendly way. In addition, we expect that this study can be applied to various base materials requiring antibacterial properties.

## Figures and Tables

**Figure 1 materials-15-00392-f001:**
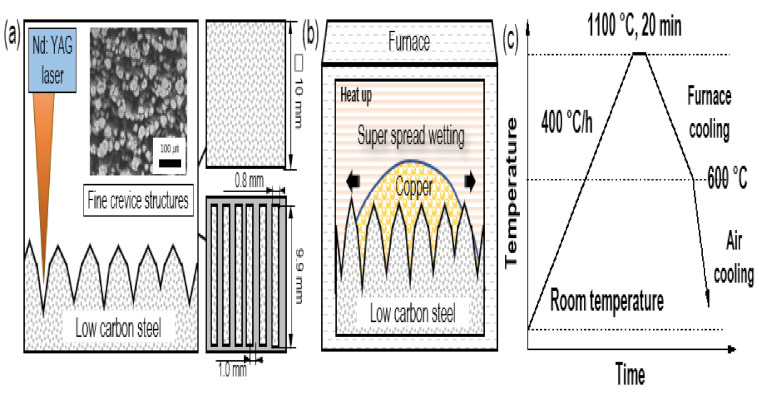
Schematic diagram of the different steps of fabricating procedure: (**a**) formation of fine crevice structure and pattern shape using Nd: YAG laser; (**b**) process for copper coating by super-spread wetting properties; (**c**) heat temperature profile for the coating process.

**Figure 2 materials-15-00392-f002:**
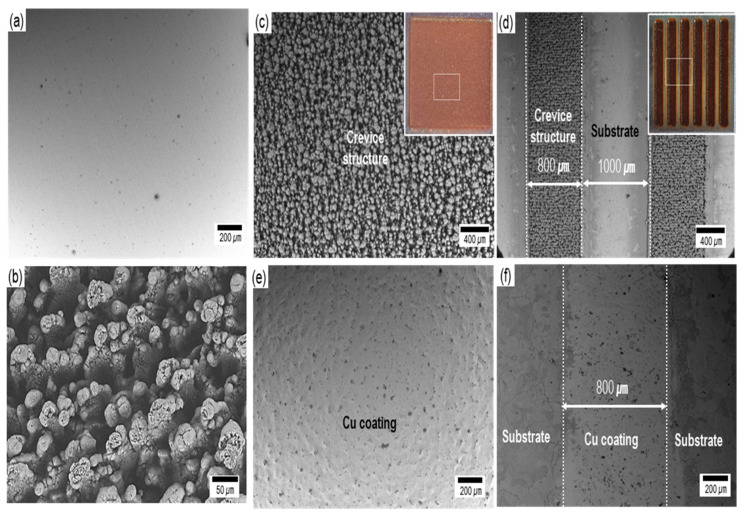
SEM images before (**a**–**d**) and after (**e**,**f**) copper coating. (**a**) as-received carbon steel substrate. (**b**) fine crevice structure by laser irradiation. Uncoated (**c**) all and (**d**) 48% of the substrate. Coated (**e**) all (**f**) 48% of the substrate.

**Figure 3 materials-15-00392-f003:**
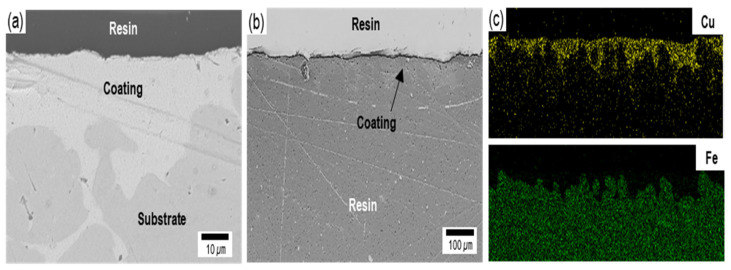
The cross-sectional images at (**a**) high- and (**b**) low-magnification with (**c**) EDS mapping.

**Figure 4 materials-15-00392-f004:**
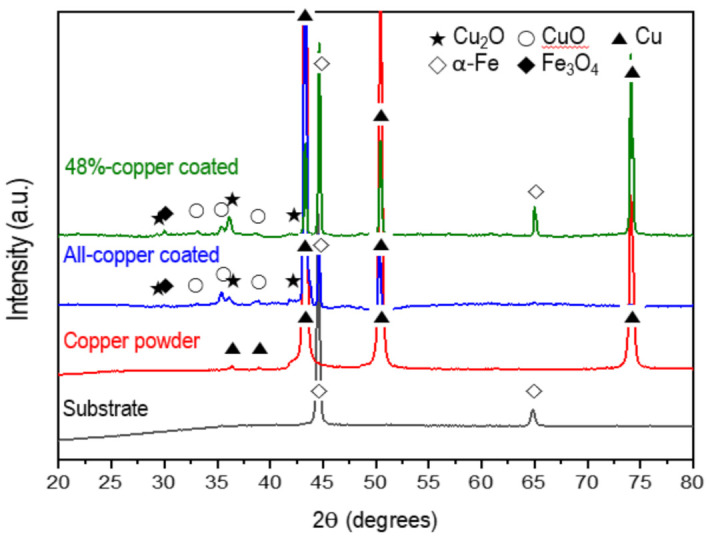
XRD patterns for specimens.

**Figure 5 materials-15-00392-f005:**
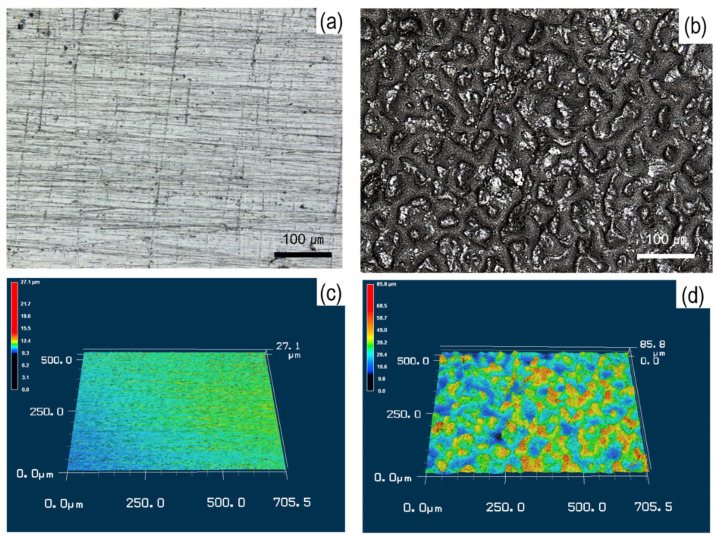
Surface and 3D images for surface topography before and after the coating process; (**a**–**c**) substrate and (**b**–**d**) coated surface.

**Figure 6 materials-15-00392-f006:**
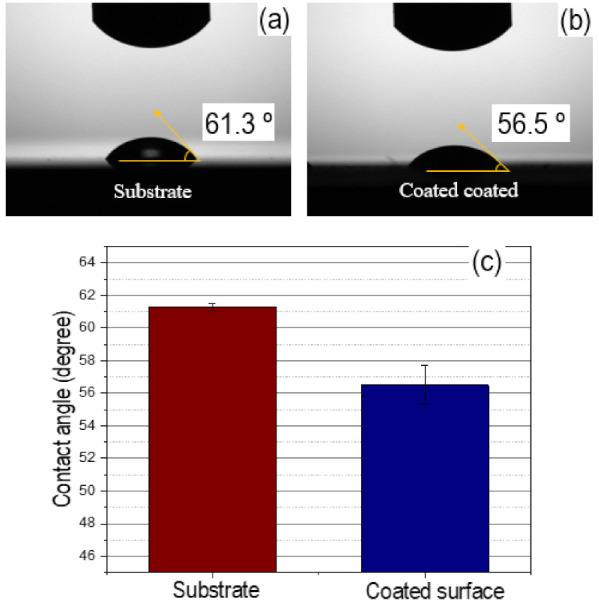
Representative images and average values for contact angle in Luria broth medium; substrate (**a**) coated surface (**b**) average values for the contact angle (**c**).

**Figure 7 materials-15-00392-f007:**
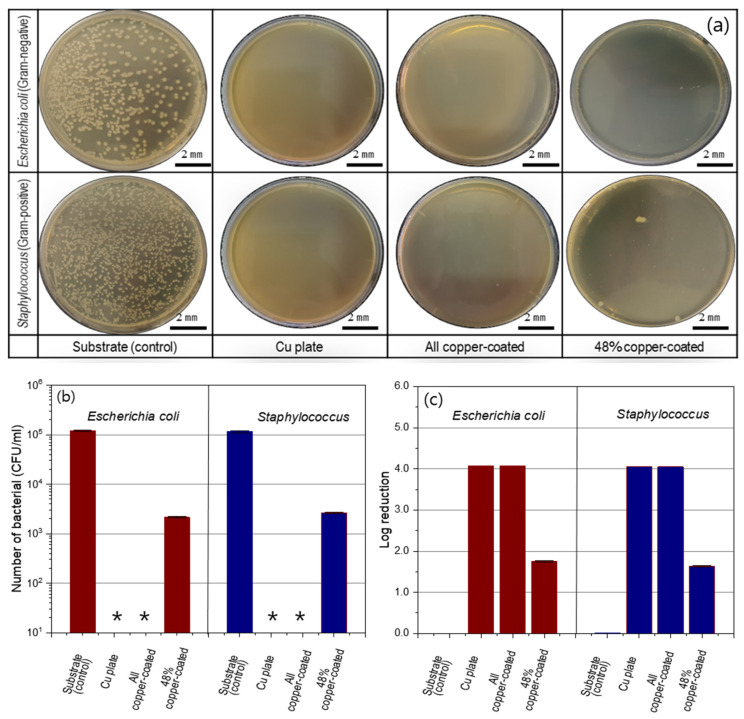
Antibacterial activity of the specimens determined by ISO 22196:2011. (**a**) Images of colonies after 24 h of incubation for *Escherichia coli* and *Staphylococcus* (dilution factor of 10^−2^), (**b**) quantified number of bacterial (* indicates that no colony was observed, limit of detection < 10 CFU/mL) and (**c**) log reduction in the different groups. The data are expressed as mean ± S.D. of triplicates.

**Figure 8 materials-15-00392-f008:**
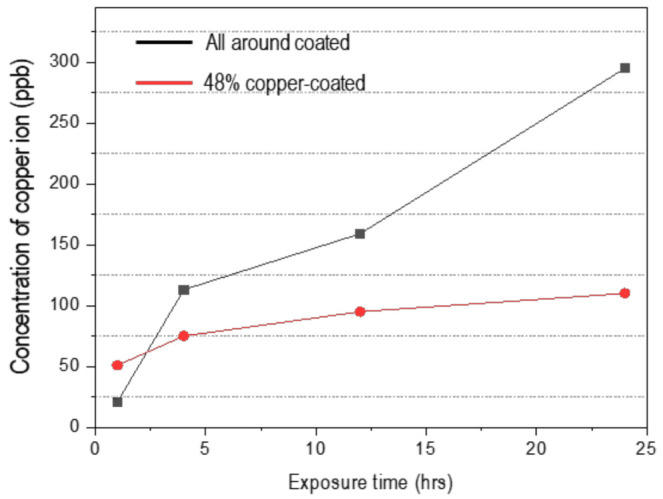
The variation in copper ion release from all-around and 48%-copper-coated in un-inoculated Luria broth medium vs. exposure time.

**Figure 9 materials-15-00392-f009:**
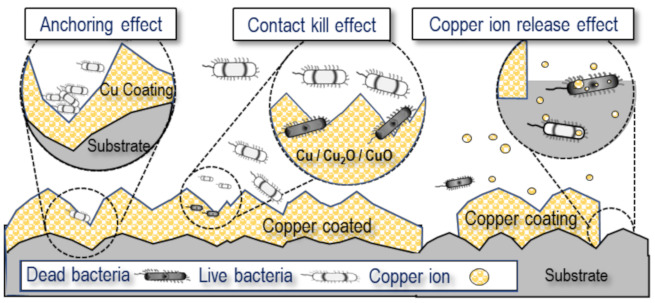
Schematic of the antimicrobial activity mechanism of copper coating specimens with super-spread wetting properties.

**Table 1 materials-15-00392-t001:** Adhesion tests according to ASTM D-3359-02 on the copper coating surface.

Sample after Adhesion Test	ASTM D3359-02 Standard	Grade
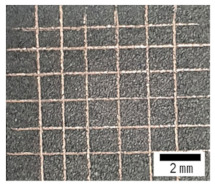	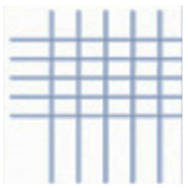	5B

**Table 2 materials-15-00392-t002:** Average roughness (R_a_), peak-to-valley roughness (R_z_) and root-mean-square roughness (R_q_) determined by 3D microscope.

	Substrate	Coated Surface
Ra (µm)	0.24 ± 0.03	6.35 ± 0.16
Rz (µm)	9.55 ± 2.35	80.57 ± 3.10
Rq (µm)	0.33 ± 0.03	7.89 ± 0.18

## Data Availability

Data will be made available upon reasonable request.

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
