# Peer review of "Copper Surface Treatment Method with Antibacterial Performance Using “Super-Spread Wetting” Properties"

_materials, 2022, doi:10.3390/ma15010392_

Round 1

Reviewer 1 Report

This work describe the preparation of copper surface treatment in order to prepare an antibacterial surface. The surface antibacterial properties  were tested against E. coli and S. aureus

The introduction is very vague, it is not clear why it is important to have a new method for copper application. It would also be important to explain to the reader where antimicrobial copper has been applied (hospitals, pipes, industries...).

The materials and methods describing the antibacterial assasy need more details. I have doubts about the cellular concentration described and I also think that some controls and clarifications are needed. Please see the attached file. In fact, the doubts about this method results also in some limitations in the interpretation of results.

The english should be significantly improved.

Author Response

19. December. 2021

Materials

Dear Ms. Reviewer,

Thank you for inviting us to submit a revised draft of our manuscript entitled, "[Copper surface treatment method with antibacterial performance using "Super-spread wetting" properties]" to [Materials]. We also appreciate the time and effort you and each of the reviewers have dedicated to providing insightful feedback on ways to strengthen our paper. Thus, it is with great pleasure that we resubmit our article for further consideration. We have incorporated changes that reflect the detailed suggestions you have graciously provided. We also hope that our edits and the responses we provide below satisfactorily address all the issues and concerns you and the reviewers have noted.

To facilitate your review of our revisions, the following is a point-by-point response to the questions and comments delivered in your comments. Please see the attachments.

Again, thank you for giving us the opportunity to strengthen our manuscript with your valuable comments and queries. We have worked hard to incorporate your feedback and hope that these revisions persuade you to accept our submission.

Sincerely yours,

Beomdeok SEO
Division of Materials and Manufacturing Science, Graduate School of Engineering, Osaka University
Address: 2-1 Yamadaoka, Suita, Osaka 565-0871, Japan
Tel/Fax: +81-6-6879-7468
E-mail: [email protected]

Authors:

Hideyuki KANEMATSU, Ph.D., Specially Appointed Professor,
Department of Materials Science and Engineering, National Institute of Technology (KOSEN), Suzuka College, Mie 510-0294, Japan
Tel/Fax: +81-59-368-1848
E-mail: [email protected]

Masashi Nakamoto, Ph.D., Assistant Professor
Division of Materials and Manufacturing Science, Graduate School of Engineering, Osaka University
Address: 2-1 Yamadaoka, Suita, Osaka 565-0871, Japan
Tel/Fax: +81-6-6879-7468
E-mail: [email protected]

Yoshitsugu MIYABAYASHI, Ph.D., Specially Appointed Professor
Graduate School of Engineering, Osaka University
Address: 2-8 Yamadaoka, Suita, Osaka 565-0871, Japan
Tel/Fax: +81-6-6879-4728
E-mail: [email protected]

Toshihiro Tanaka, Ph.D., Professor
Division of Materials and Manufacturing Science, Graduate School of Engineering, Osaka University
Address: 2-1 Yamadaoka, Suita, Osaka 565-0871, Japan
Tel/Fax: +81-6-6879-7504
E-mail: [email protected]

Reviewer 2 Report

The manuscript shows the antibacterial and wetting properties of Cu-coated steels using a "liquid copper method" processed at 1100°C which is slightly above the melting point of copper. Depending on the application, it could be an interesting alternative to coat materials with a thin layer of copper instead of using copper parts. They show the antibacterial effects on copper-coated steels. Although, the antibacterial mechanisms of Cu species are not explored in deep (Cu/ Cu2O and CuO). The manuscript is easy to read but requires some additional care in the following points:

Suggestions:

Please check English grammar and some informal language. 

Some examples:

  1. "Besides being antibacterial properties"; "Of course, the antibacterial mechanism is also discussed."
  2. Some typos: add a full stop on the sentence: "...thermodynamic calculation with the FactSage software (version 7.1)"
  3. ...Gram-negative and Gram-positive organisms, were used in the study.
  4. The following sentence  "The following protocol, called the Film Covering Method (ISO22196) was applied to 130 characterize the antibacterial properties of specimens." should be changed to:  "The Film Covering Method (ISO22196) protocol was applied to 130 characterize the antibacterial properties of specimens."

Other suggestions:

  1. Change 324.752 nm to 325 nm.
  2. Correct format/ size of Figure 7a
  3. Please clarify the sentence:

"In addition, it is very interesting that the thing the partial exposure of the substrate material on the surface exhibits antibacterial performance."

4. eq. 2, 3 and 4 are incorrect. Check if you don't confuse Cu and Cl. Please also add physical states and check the balance of the equations. Also, add references to the proposed mechanism.

What is the main question addressed by the research? Is it relevant and interesting?  They aim to understand the impact on the antibacterial activity of coated steels with copper. In my point of view, it is interesting taking into consideration that coating steels will reduce the use of 100% copper pieces.
How original is the topic? What does it add to the subject area compared with other published material? It has some degree of originality although the method to deposit copper is performed at temperatures higher than 1000ºC.  Some other techniques can be applied such as electrodeposition or PVD, but those techniques also have implementation problems. They should demonstrate that the coating is stable (adhesion).
Is the paper well written? Is the text clear and easy to read? It needs to improve English language (grammar and formality)
Are the conclusions consistent with the evidence and arguments presented? Do they address the main question posed? Yes, but in my opinion, the antibacterial mechanisms should be more explained regarding Cu species.

Author Response

(The authors gave the same response as above.)

Reviewer 3 Report

The authors present a work on producing copper coating method with antibacterial properties. I believe that this work is interesting but it requires further implementation from several points of view:

1- The abstract is unprofessionally written, it require revision in term of summarizing the method and present the main results rather than descriptive approach.

2- the keywords should be ordered alphabetically. 

3- The introduction regarding COVID-19 should be removed from the abstract and the introduction, because it doesnt related to the research, although the authors only conducted antibacterial evaluation, COVID-19 is virus, which has nothing with manuscript.

4- The introduction, especially the first part can be improved. In general, overall introduction is not informative enough and must be implemented, the authors should mention the issue with currently used approaches and the potential and novelty of their work clearly. As an example: this review https://doi.org/10.3390/antibiotics9100648 explain the possible potential of antibacterial delivery, which give potential to your work. 

5-  The methods are not clear and written in a hasty way, especially the one regarding fabrication of testing sample and antibacterial evaluation. The authors must implement and convince the referees about the measurement method presented here.

6- Overall resolution of the figures must be improved. 

7- The discussion of the results must be enriched.

8- The text of the work must be implemented with the help of a native English-speaking scientist.

Author Response

(The authors gave the same response as above.)

Reviewer 4 Report

The authors describe a study where they present an alternative copper coated method, where they apply the use of super-wetting properties of liquid copper to coat on carbon steel. The work is interesting, well structured, however there are still some issues which have to be addressed.

  • In line 172, the authors report that the coatings exhibit a homogeneous microstructure and firmly to the substrate, however the authors do not present any adhesion test to verify that there is no removal of the film.
  • The authors show that the coating is antibacterial, but I think it would enrich the study if the authors tested their cytotoxicity.

The article is well written, with a few typos that will be easily detectable after a brief review of the text.

Author Response

19. December. 2021

Materials

Dear Ms. Reviewer,

Thank you for inviting us to submit a revised draft of our manuscript entitled, "[Copper surface treatment method with antibacterial performance using "Super-spread wetting" properties]" to [Materials]. We also appreciate the time and effort you and each of the reviewers have dedicated to providing insightful feedback on ways to strengthen our paper. Thus, it is with great pleasure that we resubmit our article for further consideration. We have incorporated changes that reflect the detailed suggestions you have graciously provided. We also hope that our edits and the responses we provide below satisfactorily address all the issues and concerns you and the reviewers have noted.

To facilitate your review of our revisions, the following is a point-by-point response to the questions and comments delivered in your comments. Please see the attachments.

Again, thank you for giving us the opportunity to strengthen our manuscript with your valuable comments and queries. We have worked hard to incorporate your feedback and hope that these revisions persuade you to accept our submission.

Sincerely,

Beomdeok SEO
Division of Materials and Manufacturing Science, Graduate School of Engineering, Osaka University
Address: 2-1 Yamadaoka, Suita, Osaka 565-0871, Japan
Tel/Fax: +81-6-6879-7468
E-mail: [email protected]

Hideyuki KANEMATSU, Ph.D., Specially Appointed Professor,
Department of Materials Science and Engineering, National Institute of Technology (KOSEN), Suzuka College, Mie 510-0294, Japan
Tel/Fax: +81-59-368-1848
E-mail: [email protected]

Masashi Nakamoto, Ph.D., Assistant Professor
Division of Materials and Manufacturing Science, Graduate School of Engineering, Osaka University
Address: 2-1 Yamadaoka, Suita, Osaka 565-0871, Japan
Tel/Fax: +81-6-6879-7468
E-mail: [email protected]

Yoshitsugu MIYABAYASHI, Ph.D., Specially Appointed Professor
Graduate School of Engineering, Osaka University
Address: 2-8 Yamadaoka, Suita, Osaka 565-0871, Japan
Tel/Fax: +81-6-6879-4728
E-mail: [email protected]

Toshihiro Tanaka, Ph.D., Professor
Division of Materials and Manufacturing Science, Graduate School of Engineering, Osaka University
Address: 2-1 Yamadaoka, Suita, Osaka 565-0871, Japan
Tel/Fax: +81-6-6879-7504
E-mail: [email protected]

Round 2

Reviewer 1 Report

The authors significantly improved teh english through the entire manuscript. However, I think there are still some comments that authors did not adress in the revised form:

L.597-600 - the specimen was vortexed with 1 mL of sterilized water and tween 80 to remove unattached bacteria? Or for the detachment of bacteria from the surface and, in that way, have all detached bacteria in suspension to allow its enumeration through CFU counting in agar plates? The sentence stating the purpose is to remove any unattached bacteria does not make sense.

L.601. The cited reference [43], did not mention the use of vortex but the use of ultrassounds. Also the mentioned referrence only used S. aureus and not E. coli. So, I ask again if the authors performed some control to ensure that the procedure used for bacterial detachment for further quantification was efficient and reproducible. 

Figure 7. As I mentioned in the previous revision, this figure is not needed, it is purely ilustrative so I think that if you want to present it, please present as a supplemntary figure. In microbiology, this kind of images are not relevant, since several dilutions are performed and the images can be from different dilutions and it will not justify any difference. If you want to keep this figure, please, at least add information, in the caption, about the dilution that is represented by each picture.

L.788. As I mentioned in the previous revision, the authors should be more carefull in presenting values about CFU reduction. The first point the authors should adress is what means 100 and 98% of reduction. I understood the equation, however, the authors should present the limit of detection of the methodology. Even if the authors plate all the 1 mL of the cell suspension, there are some cells that can be missed in the detachment procedure. For this reason tehre are always a detection limit that will give information about the lower counts of CFU that can be detected.
Other question mentioned previously is the use of log reduction instead of % of reduction. In fact 98% of removal seems very effective, however, the 2% remaining are still 2000 bacterial cells. Instead of presenting the pictures of colonies, the authors should present the current CFU count for each tested situation (control and copper specimens) or the log reduction, since in microbiology it is the main way to actually undersatnd the efficacy of microbial control.

Author Response

30. December. 2021

Materials

Dear Ms. Reviewer,

We would like to express our thanks for the reviewer’s time and effort to review our manuscript. Those comments are all valuable and very helpful for revising and improving our paper. We have analyzed the comments carefully and have made corrections which we hope meet with approval.

The manuscript has been rechecked, and the necessary changes have been made in accordance with the reviewers’ suggestions. The responses to all comments have been prepared and attached herewith. Please see the attachments.

Again, thank you for your consideration.

Sincerely yours,

Beomdeok SEO
Division of Materials and Manufacturing Science, Graduate School of Engineering, Osaka University
Address: 2-1 Yamadaoka, Suita, Osaka 565-0871, Japan
Tel/Fax: +81-6-6879-7468
E-mail: [email protected]

Authors:

Hideyuki KANEMATSU, Ph.D., Specially Appointed Professor,
Department of Materials Science and Engineering, National Institute of Technology (KOSEN), Suzuka College, Mie 510-0294, Japan
Tel/Fax: +81-59-368-1848
E-mail: [email protected]

Masashi Nakamoto, Ph.D., Assistant Professor
Division of Materials and Manufacturing Science, Graduate School of Engineering, Osaka University
Address: 2-1 Yamadaoka, Suita, Osaka 565-0871, Japan
Tel/Fax: +81-6-6879-7468
E-mail: [email protected]

Yoshitsugu MIYABAYASHI, Ph.D., Specially Appointed Professor
Graduate School of Engineering, Osaka University
Address: 2-8 Yamadaoka, Suita, Osaka 565-0871, Japan
Tel/Fax: +81-6-6879-4728
E-mail: [email protected]

Toshihiro Tanaka, Ph.D., Professor
Division of Materials and Manufacturing Science, Graduate School of Engineering, Osaka University
Address: 2-1 Yamadaoka, Suita, Osaka 565-0871, Japan
Tel/Fax: +81-6-6879-7504
E-mail: [email protected]

Reviewer 3 Report

The authors have done the required recommendation, the revised manuscript can be considered for publication in Materials. 

Author Response

30. December. 2021

Materials

Dear Ms. Reviewer,

We would like to express our thanks for the reviewer’s time and effort to review our manuscript. Those comments are all valuable and very helpful for revising and improving our paper

Again, thank you for your consideration.

Sincerely yours,

Beomdeok SEO
Division of Materials and Manufacturing Science, Graduate School of Engineering, Osaka University
Address: 2-1 Yamadaoka, Suita, Osaka 565-0871, Japan
Tel/Fax: +81-6-6879-7468
E-mail: [email protected]

Authors:

Hideyuki KANEMATSU, Ph.D., Specially Appointed Professor,
Department of Materials Science and Engineering, National Institute of Technology (KOSEN), Suzuka College, Mie 510-0294, Japan
Tel/Fax: +81-59-368-1848
E-mail: [email protected]

Masashi Nakamoto, Ph.D., Assistant Professor
Division of Materials and Manufacturing Science, Graduate School of Engineering, Osaka University
Address: 2-1 Yamadaoka, Suita, Osaka 565-0871, Japan
Tel/Fax: +81-6-6879-7468
E-mail: [email protected]

Yoshitsugu MIYABAYASHI, Ph.D., Specially Appointed Professor
Graduate School of Engineering, Osaka University
Address: 2-8 Yamadaoka, Suita, Osaka 565-0871, Japan
Tel/Fax: +81-6-6879-4728
E-mail: [email protected]

Toshihiro Tanaka, Ph.D., Professor
Division of Materials and Manufacturing Science, Graduate School of Engineering, Osaka University
Address: 2-1 Yamadaoka, Suita, Osaka 565-0871, Japan
Tel/Fax: +81-6-6879-7504
E-mail: [email protected]

Reviewer 4 Report

The authors have adequately addressed my concerns and comments, and I can now recommend acceptance of this paper. 

Author Response

(The authors gave the same response as above.)
